# Determinants of *Aedes* mosquito larval ecology in a heterogeneous urban environment- a longitudinal study in Bengaluru, India

**Deepa Dharmamuthuraja[1]ᵒ, Rohini P. D.[1]ᵒ, Iswarya Lakshmi M.[1], Kavita Isvaran[2], Susanta Kumar Ghosh[3], Farah Ishtiaq[1] \***

**1** Tata Institute for Genetics and Society, New InStem Building, GKVK Campus, Bengaluru, India, **2** Centre for Ecological Sciences, Indian Institute of Science, Bengaluru, India, **3** ICMR-National Institute of Malaria Research, Bengaluru, India

ᵒ These authors contributed equally to this work.
\* farah.ishtiaq@tigs.res.in

## Abstract

### Background

*Aedes*-borne disease risk is associated with contemporary urbanization practices where city developing structures function as a catalyst for creating mosquito breeding habitats. We lack better understanding on how the links between landscape ecology and urban geography contribute to the prevalence and abundance of mosquito and pathogen spread.

### Methods

An outdoor longitudinal study in Bengaluru (Karnataka, India) was conducted between February 2021 and June 2022 to examine the effects of macrohabitat types on the diversity and distribution of larval habitats, mosquito species composition, and body size to quantify the risk of dengue outbreak in the landscape context.

### Findings

A total of 8,717 container breeding sites were inspected, of these 1,316 were wet breeding habitats. A total of 1,619 mosquito larvae representing 16 species from six macrohabitats and nine microhabitats were collected. *Aedes aegypti* and *Aedes albopictus* were the dominant species and significantly higher in artificial habitats than in natural habitats. Breeding preference ratio for *Aedes* species was high in grinding stones and storage containers. The *Aedes* infestation indices were higher than the WHO threshold and showed significant linear increase from Barren habitat to High density areas. We found *Ae. albopictus* breeding in sympatry with *Ae. aegypti* had shorter wing length.

**Data Availability Statement:** All relevant data are within the paper and its Supporting information files.

**Funding:** This research was financially supported by Tata Trusts funding to Tata Institute for Genetics and Society. The funders had no role in study design, data collection and analysis, decision to publish, or preparation of the manuscript.

**Competing interests:** The authors have declared that no competing interests exist.

## Conclusions

A large proportion of larval habitats were man-made artificial containers. Landscape ecology drives mosquito diversity and abundance even at a small spatial scale which could be affecting the localized outbreaks. Our findings showed that sampling strategies for mosquito surveillance must include urban environments with non-residential locations and dengue transmission reduction programmes should focus on 'neighbourhood surveillance' as well to prevent and control the rising threat of *Aedes*-borne diseases.

## Author summary

The quality of mosquito larval habitats (breeding sites) is one of the most important determinants of the distribution and abundance of mosquito species. Cities offer a heterogeneous landscape with a gradient of temperature, vegetation, built infrastructure (piped water access, water storage) which can vary in microclimate at fine spatial scales.

Entomological surveys are often biased towards locations or houses with high mosquito densities. Sampling strategies for mosquito surveillance must include urban environments with non-residential locations. Understanding the linkages between environmental conditions (e.g., hydrology, microclimate), land use, climate change, increasing urbanization are some of the key factors modulating the mosquito life-history traits which influence epidemiologically relevant behaviors and their ability to transmit diseases. Our longitudinal study shows that a combination of manmade larval habitats and landscape ecology drives mosquito diversity and abundance even at a small spatial scale which could be affecting the incipient disease outbreaks. From science to policy perspective, this is the first comprehensive study from Bengaluru, India which shows that sampling strategies for mosquito surveillance must include urban environments with non-residential locations. We demonstrate that dengue transmission reduction programmes should focus on 'neighbourhood surveillance' as well to prevent and control the rising threat of *Aedes*-borne diseases.

## Introduction

Mosquito-borne diseases, particularly dengue, chikungunya transmitted by *Aedes* mosquitoes are rising worldwide and have been a critical public health issue. In urban ecosystems, *Aedes*-borne disease risk is associated with contemporary urbanization practices where city developing structures functions as a catalyst for creating breeding habitats for two epidemiologically important mosquito species—*Aedes aegypti* [= *Stegomyia aegypti*] and *Aedes albopictus* [= *Stegomyia albopicta*] [1]. Mosquitoes as an obligatory host of many parasites can adapt to a wide range of ecological conditions. From a microhabitat perspective, the quality of mosquito larval habitats (breeding sites) is one of the most important determinants of the distribution and abundance of mosquito species [2–6]. *Aedes* species are often associated with a specific type of breeding site, from temporary, ephemeral habitats such as water-filled leaf axils, coconut shells, tree holes to manmade habitats such as ground pools, water storing containers, pots, and tyres etc. [5]. In an ecological context, the mosquito breeding success is tightly linked with the stability of aquatic habitats and is exquisitely dependent on temperature, humidity, and rainfall [7,8]. Many biotic (predators, organic matter, larval density, interspecific competition) and abiotic (temperature, rainfall, and humidity) factors influence the larval and pupa population

dynamics and determines the life-history traits (longevity, fertility, body size, and immune function). For example, predation at larval stages has been identified as an important evolutionary force driving the habitat segregation and niche partitioning in mosquito species [3]. Numerous studies have demonstrated that wing length is a good proxy for fitness and survival of mosquito species [9]. In addition, biotic and abiotic conditions of larval habitats determine the abundance and body size of emerging adult mosquitoes [10]. Larger mosquito size is positively associated with survival, blood feeding frequency, which is likely to increase disease transmission [2,11–15].

Globally, the number of dengue cases has increased 30-fold over the past five decades [16]. The first case of dengue-like illness was reported in Chennai in 1780 and first confirmed dengue viral infection occurred in Calcutta (now Kolkata) in 1963–1964 [17]. Since 1968, many parts of northern India (Kanpur, Delhi) have experienced dengue outbreaks. In the early 2000s, dengue was endemic in a few southern (Maharashtra, Karnataka, Tamil Nadu and Pondicherry) and northern states (Delhi, Rajasthan, Haryana, Punjab and Chandigarh [18]. One of the limitations for dengue control has been the lack of structured mosquito surveillance, diagnostics and awareness and collective efforts which has led to high number of cases [19]. Among arboviruses, dengue virus diagnosis relies on the detection of the virus or antibodies (IgM, IgG, and NS1 antigen) directed against the virus in the blood [20]. In addition, the sensitivity and specificity of these tests underestimate the 'true' burden of the disease which poses a logistical challenge and limited compliance to invasive sampling procedures [21].

Dengue is an annual epidemic in India. In 2019, dengue burden in India peaked at about 1,57,315 cases and Karnataka recorded 16,986 cases. Of these Bengaluru contributed ~50% (9,029) of dengue cases (National Centre for Vector Borne Diseases Control; NCVBDC). Like many cities, in Bengaluru, *Aedes* population surveillance primarily involves indoor larval surveillance as per WHO protocols to measure house index, container index and Breteau index to quantify the disease risk in a specific residential area [22]. Source reduction (emptying water holding containers), anti-larval spraying and providing health education/awareness are the main intervention strategies for *Aedes* control (NCVBDC). The areas with highest house indices and larval counts are considered as productive. These indices record relative larval abundance in a locality for a specific period with no correlation with adult abundance and without regard for seasonal fluctuation in larval abundance. Furthermore, entomological surveys are often biased towards locations or houses with high mosquito densities or disease outbreaks [22]. Sampling strategies for mosquito surveillance must include urban environments with non-residential locations [23,24]. Chen *et al.* [25] analysis of weekly data during COVID-19 pandemic across southeast Asia and Latin America showed a drastic decline in dengue cases due to lockdowns and restricted movements of people. These findings further highlighted the dengue transmission reduction programmes should focus on 'neighbourhood surveillance' as there is no relationship between the Breteau and house indices of *Aedes* mosquitoes with dengue fever outbreaks [26].

Cities offer a heterogeneous landscape with a gradient of temperature, vegetation, built infrastructure (piped water access, water storage) which can vary in microclimate at fine spatial scales less than 1km x 1km [27,28]. These differences further affect the vector abundance, fitness traits and virus transmission dynamics at a microhabitat level. Cities provide a diverse gradient of larval habitats such as construction sites [29], leaking connections [30] among other aquatic habitats created by anthropogenic land use modifications (e.g., bromeliads, Colocasia plants, buckets, plastic containers etc) which are positively associated with the abundance of *Aedes* species. Understanding the linkages between environmental conditions (e.g., hydrology, microclimate), land use, climate change, increasing urbanization are some of the key factors modulating the mosquito life-history traits which influence epidemiologically

relevant behaviors and their ability to transmit diseases. For example, with changing water infrastructure networks in cities like Bengaluru, freshwater wells and lakes were replaced by formal water connections, the prime larval habitat of urban malaria vector *Anopheles stephensi*, is now prevalent in water storage (cement/plastic tanks, plastic drums) and often shared niche with *Aedes* species such as discarded tyres. With increasing urbanization and availability of man-made habitat has led to establishment of *Aedes* species. Similarly, in Ahmedabad (Gujarat), more malaria cases were associated with the areas with low density of formal water connections [30]. Larval habitat characteristics such as pH, temperature, salinity predict the presence of mosquito larvae and rearing environment has impact on body size of females, daily survival rate and susceptibility to arboviral infections [31]. We conducted a longitudinal study to understand how urbanization affects mosquito ecology and how mosquito species diversity and abundance changes across macro and microhabitats in an urban environment. We are specifically interested in understanding:

1. What types of habitats (microhabitat and macrohabitat) and season drives larval presence and species diversity?

2. How do *Aedes* indices change by season and macrohabitat types?

3. How do abiotic factors predict the abundance of mosquito larvae across an urbanization gradient?

4. How do abiotic factors and urbanization drive niche conservatism/habitat segregation in *Aedes* species?

5. How does wing length vary between interspecific and intraspecific environments?

## Methods

Bengaluru (12°58′44″N, 77°35′30″E), Karnataka, India, is a densely populated (~11 million inhabitants) city at an elevation of 900 m above mean sea level, with a mosaic of urbanized areas to plantation, barren areas representing a diverse range of land use classes (Fig 1). The study was conducted between February 2021 and June 2022. Bengaluru experiences a semi-arid climate with temperature between 18°C to 34°C, and southwest monsoon season (June-September) with normal annual rainfall of 820mm which accounts for nearly 60% of the rainfall [32]. The retreating, northeast monsoon also brings rain from October to December. The dry period extends from January to May, although convectional thunderstorms occur from March through May. Typically, January and February receive almost no rain.

Our goal was to quantify seasonal variation in the *Aedes* larval prevalence, degree of breeding habitat utilization (microhabitats and macrohabitats) and niche conservatism in mosquito species. To examine the seasonality in the larval habitat prevalence and abundance of mosquito species, we divided the year into biologically meaningful four seasons: dry (January-March), pre-monsoon (April-June), monsoon (July-September), post-monsoon (October-December). Due to unprecedented COVID-19 waves (Delta and Omicron), the fieldwork was staggered in two years (2021–2022) to capture the seasonal data. We defined six macrohabitats with eight replicates in a 100m x 100m grid in following categories: barren lands, lakes (lake surroundings), plantations, high density areas (e.g., >75% grid coverage by households), medium density areas (> 50% grid coverage by households), low density areas (> 25% grid coverage by households). These macrohabitat grids were selected using Google Earth Pro (v.7.3) and were subsequently verified in the field.

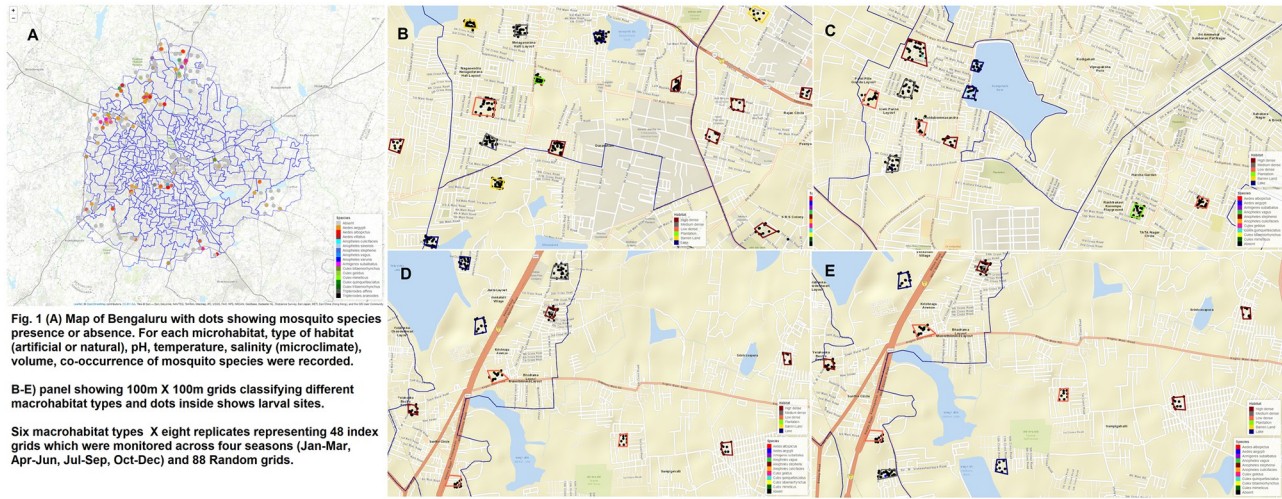

Fig. 1 (A) Map of Bengaluru with dots showing mosquito species presence or absence. For each microhabitat, type of habitat (artificial or natural), pH, temperature, salinity (microclimate), volume, co-occurrence of mosquito species were recorded.

B-E) panel showing 100m X 100m grids classifying different macrohabitat types and dots inside shows larval sites.

Six macrohabitat types X eight replicates representing 48 index grids which were monitored across four seasons (Jan-Mar, Apr-Jun, Jul-Sep, Oct-Dec) and 88 Random grids.

**Fig 1. Map of Bengaluru showing grid locations by macrohabitat types and larval sites by mosquito species.** This map was created using the *Leaflet* package and OpenStreetMap (https://www.openstreetmap.org/about).

A total of 48 index grids (fixed) were selected with eight replicates of each macrohabitat type (low density, medium density, high density, plantation, lake area, and barren land) which were surveyed in each season. In addition, we selected 98 random grids representing a macrohabitat type which were surveyed in two seasons (April-June and July-September), which allowed us to sample mosquitoes in varied ecological niches in the city while surveying fixed grids on a seasonal basis. The minimum distance between each grid (index and random) was approximately 300m-500m to maximum 1km. We deployed ten iButton data loggers to record temperature and relative humidity in index grids representing each macrohabitat type. The loggers were deployed in completely shaded area and were set up to record at 30 min interval round -the- clock basis for the duration of the study.

## Mosquito larval habitat sampling

A grid representing a macrohabitat was sampled within 1 day. Each grid was surveyed for the standing water i.e., microhabitat (breeding source or larval habitat) categorised as natural (e.g., plant axils, tree holes, coconut shells) and artificial (e.g., discarded plastic containers, discarded grinding stone, discarded tyres, plant pots, collection plates, storage containers, stagnant water). Each microhabitat was visually inspected for the presence of mosquito larvae. A microhabitat was determined as 'positive' if a larva was recorded. For each sampling location, geo-coordinates, microclimate variables such as pH, temperature using HM Digital pH meter and salinity using salinity refractometer were recorded. We did not use salinity in subsequent analysis due to zero values across all habitats. Water volume of each habitat was measured using meter stick (>10L) or by transferring it a graduated beaker. We used *leaflet* package, an open-source JavaScript library for interactive maps to map larval sites.

Breeding preference ratio (BPR) for *Aedes* mosquitoes across all breeding sources was calculated to assess the preference for available breeding habitats. BPR was calculated using the ratio of number of containers infested with *Aedes* larvae to the number of water-holding containers examined [33–35].

The mosquito larvae samples were collected and brought in collection containers for rearing to mosquitoes in the insectary at the Tata Institute for Genetics and Society. Larvae were

reared separated by collection source where they were placed in 50-100mL of collection water and provided with fish food. Larvae and pupae were reared to adulthood at 28±2˚C and 75± 5% relative humidity and 12:12 hour light-dark photocycle. Once emerged, mosquitoes were frozen at -20˚C, sorted by gender and identified to species following Barraud *et al.* [36].

## Adult mosquito sampling

We conducted *Aedes* adult sampling in four grids in high density areas (due to high dengue incidence) in each season. Because *Aedes albopictus* and *Aedes aegypti* are day-biting mosquitoes, Biogents (BG) sentinel trap using battery operated fan was deployed for 72 hours for one sampling period. Mosquito traps were baited with a BG-Lure cartridge (Biogents) and an octenol (1-octen-3-ol) lure inside the trap. Traps were placed under the cover outside a house premises to increase catch rates. The battery was changed in the morning for each trap day and catch bags were collected and replaced with a new catch bag to reduce destruction of samples. Adult collections were brought to the laboratory and stored at –20˚C. Mosquitoes were sorted by sex and species following Barraud *et al.* [36]. Adult mosquito abundance for three trap days were combined to calculate the total abundance for that sampling period.

## *Aedes* egg sampling

Ovitraps were deployed in four high density grids (due to high dengue incidence). Each ovitrap was a simple black plastic container of approximately 500 ml capacity and a diameter of 5 cm and 10 cm height. Each container was lined with an oviposition strip, lining the inner wall of each trap, which was withdrawn after exposure to oviposition for 5 consecutive days. These traps were deployed on the ground simultaneously in a separate high dense grid to avoid compromising the adult sampling. In each grid, a group of five ovitraps in five replicates i.e., 25 ovitraps. The grid was considered positive if eggs were recorded in ovitraps. The eggs were counted using stereomicroscope and ovitrap index were calculated using the percentage of positive ovitraps against the total number of ovitraps recovered from each site [37].

## Wing length as an indicator of body size

Wing length of emerged adult mosquito was recorded. The newly emerged mosquitoes were identified and kept in -20˚C and wing lengths was measured within 2 days of emergence. Both right- and left-wing length was measured for male and female *Aedes* mosquitoes using a microscope and ocular micrometer as the distance from the axillary incision to the apical margin, excluding the fringe scales [38]. We used Leica stereomicroscope with the LAS X (Leica application suite x) software platform to capture and quantify the specimen. Wing length for the field caught *Aedes* mosquito in high density urban areas was also measured in the similar way. We considered right wing length of females in the subsequent analysis.

## Data analyses

Data were analyzed in R v.2023.6.2 [39]. Exploratory analyses were first performed to identify environmental differences between macrohabitats, as well as variations in microclimate characteristics according to larval habitat categories (S1 Data).

## Macrohabitat type and season

For index grids, (6 habitat types x 8 replicates x 4 seasons), a binary logistic regression model was used to assess the relationship between larval habitat characteristics and the presence of immature stages. We used three generalised linear mixed effect model (GLMMs, function

glmer in lme4; Bates *et al.* [40] where larval presence/absence was response variable with season, macrohabitats, microhabitats, pH and temperature as the fixed effects and grid as random effect. Similar models were considered for assessing the effect of these variables on the prevalence of *Ae. aegypti* and *Ae. albopictus*. This analysis was also conducted separately for all (index + random) grids. The significance of fixed effects was evaluated with Wald's $\chi2$-tests and β estimates with *p* values are reported [41]. Subsequently, all grids were considered together (see results).

For all grids together, we estimated diversity indices for microhabitats (artificial vs natural) and individual rarefaction curves. The total richness by each habitat type was estimated by abundance based Chao1 estimator using **iNEXT** package [42]. We constructed a grid x species mosquito abundance matrix to understand the temporal and habitat segregation of species using two measures of species diversity: (a) species richness, defined as total number of species sampled; (b) the Shannon-Wiener index, a measure of species diversity weighted by relative abundance [43]. One-way analysis of variance (ANOVA) was used to test difference in species diversity across habitat types (Barren Land < Lake < Plantation < Low density < Medium density < High density) and season (Jan-Mar < Apr-Jun < Jul-Sep < Oct-Dec)

We used analysis of variance to test whether temperature varied significantly between air temperature recorded by data logger and water temperature of microhabitat. Tukey's Honest Significant Difference post-hoc test was used to determine the significance of a pairwise comparison.

## How do *Aedes* indices change by season and macrohabitat type?

The larval indices for *Aedes aegypti* and *Aedes albopictus* were calculated to assess the risk of dengue by season and habitat type. We used thresholds indicating dengue outbreak risk based on traditional House and the Breteau indices (HI = 1%, BI = 5). These indices are analogous to traditional Container, House (or Premise), and Breteau indices [37] but considers non-household locations in their calculation [22]:

*Container index*: Number of habitats positive for *Aedes* larvae and/or pupae x 100 / total potential containers inspected in each habitat.

*Location index*: Number of locations positive for *Aedes* larvae and/or pupae x 100 /total number of locations inspected in each habitat multiplied by 100.

*Breteau location index*: Number of habitats positive for *Aedes* larvae and/or pupae x 100 / total number of locations inspected in each habitat [22].

*Pupal index*: Number of pupae collected of total potential containers inspected in each habitat.

We used generalised linear model (GLM) to tease apart the effect of macrohabitat type and season on larval indices. We fitted three separate Poisson regression models using location index, container index and Breteau location index. The low sample size prevented inclusion of interactive terms or random effects in the model. In addition, to account for zero values of some larval indices, a constant value of 1 was added to fit the GLM model with a log-link function. This did not affect the mean and variance of that variable.

## How do abiotic factors predict the abundance of mosquito larvae across macro and microhabitats?

To determine effects of larval habitat characteristics on larval abundance, we used generalized linear mixed effects models (GLMMs) with a negative binomial link because our count data were over dispersed. We modelled the effect of microhabitat, habitat type (macrohabitat) and

microclimate variables (pH and temperature), volume, sympatric (co-occurring) species (yes/no), and season on the larval abundance (count) and including grid as random effect. The microclimate variables (pH and temperature) were fitted using a basis-spline (B-spline) function to allow for non-linear relationships with larval abundance. A candidate set of 128 possible models were fitted separately for total larval abundance, *Ae. aegypti* and *Ae. albopictus* larval abundance. GLMMs were fitted using the *glmmTMB* package [44]. Scaled residuals of the models were inspected for overdispersion and uniformity using the *DHARMa* package [45]. We assessed model support based on ΔAICc, the AIC value corrected for small sample sizes [46]. Models with a ΔAICc value < 2 were considered as having greatest support, with the awareness that parameters from models with a K-value greater than that of the top supported model may not be truly informative [47]. Nevertheless, parameter estimates from all models ranking within 2 ΔAICc were averaged using the R package "AICcmodavg" to investigate the relative significance of parameters within this set of top supported models. We then calculated 95% conditional confidence intervals of each parameter or model-averaged parameter and identified those that did not overlap zero as important predictors, recognizing that parameters not occurring in the top model may hold questionable importance.

### How do abiotic factors and urbanization drive niche conservatism in *Aedes* species?

We used a GLMM model where the presence of *Ae. aegypti* species was assessed as a function of presence of *Ae. albopictus*, temperature, pH (microclimate), season, and habitat type (natural and artificial) with binomial distribution and grid as random effect. The sample size of other mosquito species was small (<70) to be considered for this analysis. An odds ratio value of one indicates species are associated randomly, whereas odds ratio values of greater than one or less than one indicates a positive or negative association, respectively. All analyses were carried out in *lme4* package [40].

### How does wing length vary between interspecific and intraspecific environments?

The wing lengths of *Ae. aegypti* and *Ae. albopictus* were compared between either species originating in larval habitats with and without *Aedes* species. Similar comparisons between larval and adult caught mosquito wing lengths were tested by Kruskal Wallis test as wing length was not considered normally distributed.

We fitted Linear Mixed Models (LMM) using Gaussian distribution to explore the association between the wing lengths (as response variable) of the *Aedes* mosquitoes with microclimate (temperature, pH), larval abundance, microhabitat, macrohabitat and season as fixed effects and grids were added as random effect. These models were fitted separately for *Ae. aegypti* and *Ae. albopictus*.

## Results

In this neighbourhood (non residential outdoor) surveillance, 242 grids were sampled for mosquito larval habitats. Of these a total of 106 grids (48 index + 58 random) were sampled from April to June, and 88 grids (48 index + 40 random) were sampled from July to September. The third wave of COVID-19 disrupted the field work during October-December which restricted sampling to 48 index grids. A total of 8,717 container breeding sites were inspected, of these 1316 were potential wet breeding habitat. A total of 1,619 mosquito larvae were collected from six macrohabitats and nine microhabitats. Of these 1,290 mosquitoes emerged

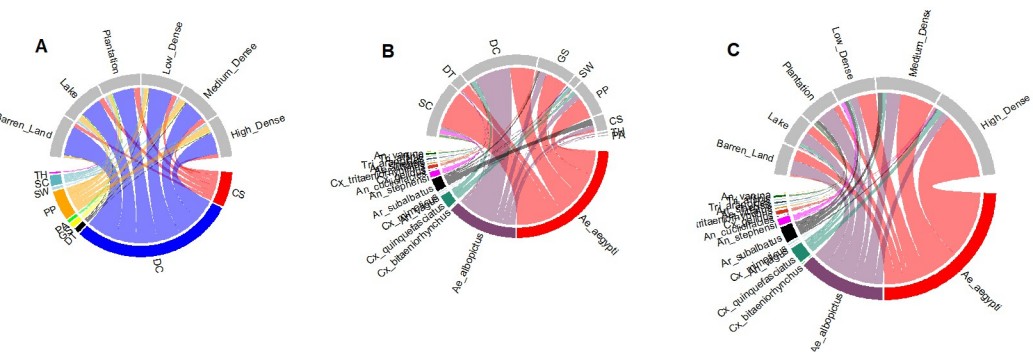

**Fig 2. Circos plots showing macrohabitat and microhabitat niche segregation by mosquito species.** A: Proportion of microhabitat recorded in each macrohabitat type; B: Proportion of mosquito larval sites by species recorded in each microhabitat, the larval prevalence varied significantly by microhabitat; C: Proportion of mosquito larval sites by species recorded in each macrohabitat, the larval prevalence had no significant effect of macrohabitat. CS = Coconut shells; DC = Discarded containers; PP = Plastic pots; DT = Discarded tyres; TH: Tree holes; PA = Plant axils; SC = Storage containers; GS = Grinding stones.

comprising 16 species from five genera. *Ae. aegypti* was the most dominant species 707 (55%), followed by *Ae. albopictus* 367 (28%), *Culex quinquefasciatus* 69 (5%), *Armigeres subalbatus* 69 (5%), *Anopheles stephensi* 29 (2%), *Culex gelidus* (1%), *Tripteroides affinis* (1%). Nine mosquito species- *An. vagus*, *An. sinensis*, *An. culicifacies*, *Cx. bitaeniorhynchus*, *Cx. tritaeniorhynchus*, *Cx. mimeticus*, *Tripteroides aranoides*, *Ae. vittatus*, and *An. varuna* were sampled in less than 1%. The most abundant microhabitat type was discarded containers followed by coconut shells and plant pots which were recorded across all macrohabitats, and the least abundant were plant axils and tree holes (Fig 2A).

For both *Aedes* species, the BPR was significantly different in microhabitats (Wald's $\chi^2$ = 18.47, df = 7, $p$<0.01) with highest in discarded grinding stones ($\beta$ = 1.75, $p$<0.02), and storage containers ($\beta$ = 1.55, $p$<0.04; Fig 3). However, the BPR showed no significant variation by season (Wald's $\chi^2$ = 4.7, df = 3, $p$>0.19).

The *Stegomyia* indices were higher than the WHO threshold values (Breteau index: >5%) suggesting a high level of *Ae. aegypti* and *Ae. albopictus* infestation (S1 Table). The location index, container index and Breteau location index showed significant linear increase from Barren land to High density grids (Wald's $\chi^2$ = 28.10, df = 5, $p$<0.001) and a quadratic increase during July-September followed by a decrease from October to December (Wald's $\chi^2$ = 20.20, df = 3, $p$<0.001; Fig 4). We found no significant difference in pupal index by habitat type or season. However, ovitrap index was higher in July-September (3.17) than April-June (0.41) and October-December (0.78).

In 146 BG trap nights in four high density grids between April and December 1609 mosquitoes representing five species were collected. In contrast to larval sampling, the adult collections were dominated by *Cx. quinquefasciatus* 1006 (63%), *Ar. subalbatus* 456 (28%) followed by *Ae. aegypti* 109 (7%) and *Ae. albopictus* 24 (1%). *An. stephensi* was only sampled twice. In general, males were significantly higher in abundance than females.

### Larval prevalence across seasons and habitats

We surveyed a total of 8,717 potential mosquito larval sites representing nine microhabitats (breeding sources) from February 2021 to June 2022. Out of these 7,454 (86%) were dry (potential) habitats and 1263 (14%) were wet habitats (S2 Table). In general, the prevalence of

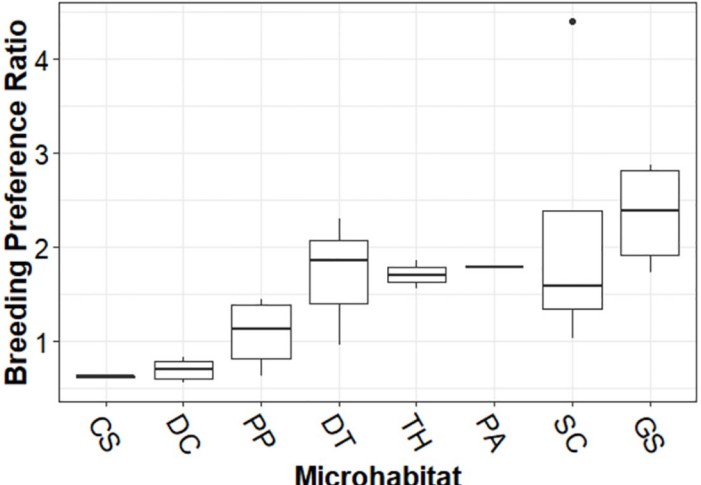

**Fig 3. Breeding preference ratio of *Aedes* species across microhabitats.** CS = Coconut shells; DC = Discarded containers; PP = Plastic pots; DT = Discarded tyres; TH: Tree holes; PA = Plant axils; SC = Storage containers; GS = Grinding stones.

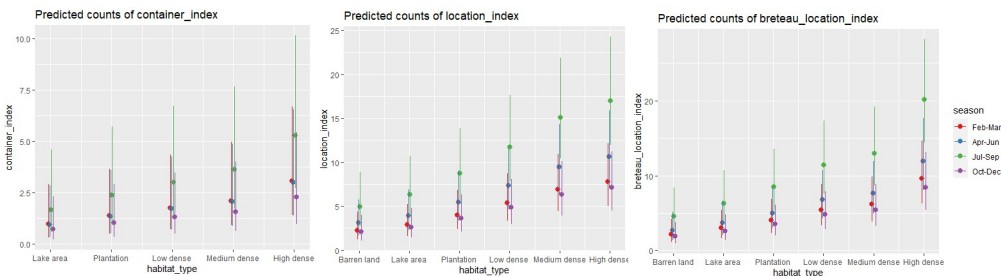

**Fig 4. *Ae. aegypti* and *Ae. albopictus* infestation indices across a macrohabitat gradient.**

wet habitat was significantly higher in index grids than in random grids (Wald's $\chi^2$ = 98.25, $p$<0.001) and this pattern remained consistent even after combining all the grids. The wet larval habitat was significantly high in artificial breeding sources (Wald's $\chi^2$ = 57.11, $p$<0.001), and this corresponded well with the finding that larval prevalence was significantly higher in artificial habitats than in natural habitats (Wald's $\chi^2$ = 7.11, $p$<0.001). There was a stark seasonal difference in the prevalence of wet habitat (Wald's $\chi^2$ = 43.11, df = 3, $p$<0.001). The prevalence of wet habitat was significantly lower in February-March (β = -1.73, $p$<0.001), and October-December (β = -1.29, $p$<0.001). However, in monsoon season, July-September (β = 0.53, $p$<0.03) showed significantly high wet habitat prevalence (Fig 5). Microclimate variables such as pH (Wald's $\chi^2$ = 1.13, $p$>0.28) and temperature (Wald's $\chi^2$ = 0.13, $p$>0.71) showed no significant difference between the positive and negative larval habitats. There was a significant variation in mean daily temperature across months but not across macrohabitat types (S1 Fig). Mean daily temperature were coolest in December (22°C) and warmest in April-May (~30°C). There was a stark difference in water temperature among microhabitat types (F = 9.20, df = 7, $p$<0.001; Tukey HSD post-hoc tests; S3 Table). The air temperature was significantly lower than water temperature of larval habitats—discarded containers (F = 28.52, $p$<0.0001),

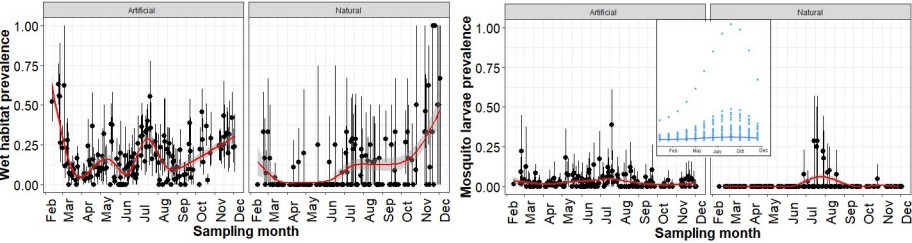

**Fig 5.** Left panel: The wet habitat prevalence, Right Panel: Mosquito larvae prevalence by month. Inset figure shows peak in dengue cases (aggregated from 2016–2019) as reported by Bruhat Bengaluru Mahanagara Palike.

discarded grinding stones ($F$ = 12.75, $p$<0.001), stagnant water ($F$ = 9.51, $p$<0.002), storage containers ($F$ = 11.82, $p$<0.0001), and pots ($F$ = 11.68, $p$<0.001). Water temperature in larval habitats such as coconut shells ($F$ = 2.28, $p$>0.14), plant axils ($F$ = 1.69, $p$>0.22), and tree holes ($F$ = 0.07, $p$>0.78) showed no significant difference with air temperature.

The top models explaining mosquito larval prevalence included season + microhabitat type + temperature + volume (Fig 6 and S4 Table). The prevalence of mosquito larvae showed significant difference across season (Wald's $\chi^2$ = 77.30, df = 3, $p$<0.001) by following a similar seasonal trend as wet habitat prevalence with significantly high prevalence in April to June ($\beta$ = 0.60, $p$<0.02) and July to September ($\beta$ = 0.51, $p$<0.05). However, October to December ($\beta$ = -1.49, $p$<0.001) showed significantly low larval prevalence. While mosquito larval prevalence showed no effect of pH (Wald's $\chi^2$ = 0.96, $p$>0.32), or macrohabitat type (Wald's $\chi^2$ = 2.20, df = 5, $p$>0.82), the microhabitats showed significant effect on the larval presence (Wald's $\chi^2$ = 60.63, df = 7, $p$<0.006). Human associated microhabitats, such as discarded grinding stones ($\beta$ = 1.33, $p$<0.002) supported significantly high larvae prevalence followed by discarded tyres ($\beta$ = 0.95, $p$<0.05; Fig 2B). There was a decrease in larval prevalence with increase in temperature (Wald's $\chi^2$ = 6.16, $p$<0.01).

The patterns in larval prevalence were primarily driven by *Ae. aegypti* and *Ae. albopictus* with similar ecology, albeit with contrasting seasonal and microhabitat usage (Fig 6 and S5 and S6 Tables). *Ae. aegypti* showed a significantly low prevalence in February and March ($\beta$ = -2.53, $p$<0.0001) and from October to December ($\beta$ = -0.98, $p$<0.01). However, no significant change in prevalence from April to June and July to September. *Ae. albopictus* showed similar levels of prevalence in February and March ($\beta$ = -4.57, $p$<0.001) with a spike in

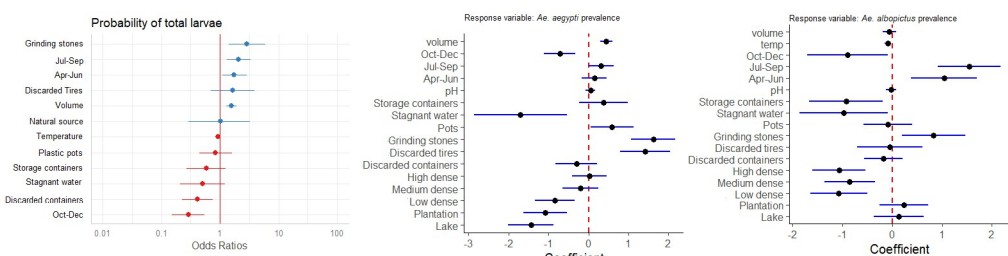

**Fig 6. Model summary of parameter estimates from the best-fit generalized linear mixed-effects model predicting larval prevalence, based on a candidate set model combinations of the variables: macrohabitat, microhabitat, pH, temperature, volume, season and co-occurring species.** The models also include the random effects of grid.

prevalence from July to September ($\beta$ = 1.50, $p$<0.01). *Ae. aegypti* showed a positive association with coconut shells ($\beta$ = 0.06, $p$<0.03) whereas the prevalence of *Ae. albopictus* was negatively associated ($\beta$ = -3.59, $p$<0.001) with coconut shells as a larval habitat. *Ae. aegypti* had significantly positive association with discarded grinding stones ($\beta$ = 1.55, $p$<0.001), and storage containers ($\beta$ = 1.31, $p$<0.01). *Ae. albopictus* showed a positive association with discarded grinding stones ($\beta$ = 1.47, $p$<0.02). *Ae. aegypti* showed significantly less prevalence in lake areas ($\beta$ = -1.21, $p$<0.03) whereas *Ae. albopictus* showed a significant negative association with high density habitat ($\beta$ = -1.14, $p$<0.03) as well as temperature ($\beta$ = -0.09, $p$<0.03).

## Mosquito species richness across seasons and habitats

For all grids, mosquito species richness (Chao1 estimator and confidence interval) was higher in artificial habitats (Observed = 16, Chao1 = 18.24; CI:16.26–35.02) than in natural habitats (Observed = 3, Chao1 = 3; CI: 3–3.01). Among artificial habitats, storage containers harboured highest mosquito richness (Observed = 11; Chao1 = 11.64; CI: 11.05–18.78) followed by stagnant water (Observed = 9; Chao1 = 16.78; CI: 9.96–18.70) and discarded containers (Observed = 6; Chao1 = 6; CI: 6–7.48). Our individual-based rarefaction curves indicated that we sampled expected number of species in each habitat except mosquito community in stagnant water remained under sampled (Fig 7).

For combined grids (index and random) analysis, both species richness ($F$ = 2.31, $p$<0.04) and Shannon diversity ($F$ = 2.21, $p$<0.05) showed a significant negative quadratic coefficient indicating that mosquito diversity indices do not decrease linearly but show a hump-shaped pattern with macrohabitat type i.e., lowest in Barren land and peaks in plantation and then declines in high dense grids. Similarly, both species richness ($F$ = 3.79, $p$<0.01) and Shannon diversity ($F$ = 3.04, $p$<0.03) showed a significant decline with a cubic coefficient from October to December (Fig 8).

For index grids, there was no significant association between richness and species diversity with macrohabitat type. However, only species richness ($F$ = 2.54, $p$<0.05) showed a marginally significant decrease from July to December whereas species diversity showed no change with season ($F$ = 2.08, $p$>0.10).

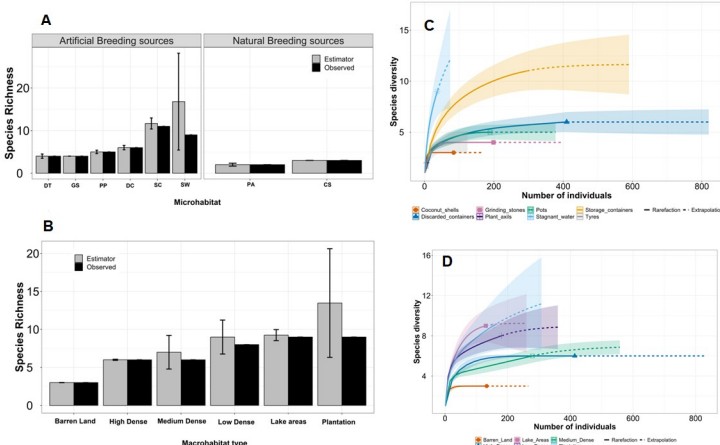

**Fig 7. Observed and Chao 1 estimated mosquito richness by A) microhabitat and B) macrohabitats and species diversity in C) micro and D) macrohabitats.**

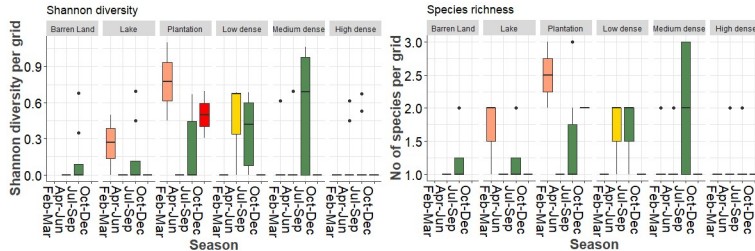

**Fig 8. Mosquito species richness and diversity across season by macrohabitat types.**

## Abundance of mosquito larvae across micro and macrohabitats

The top model explaining total larval abundance included volume + season + co-occurring species + a random effect of grid (weight = 0.77; S7 Table). The total larval abundance was negatively related to volume (effect size: -0.17; 95% CI: -0.27 to -0.07). In contrast to total larval prevalence, the larval abundance showed a gradual decrease from February to March and October-December. However, the total larval abundance was positively related to presence of sympatric species (effect size: 0.36; 95% CI: 0.11 to 0.60) (Fig 9).

For *Ae. aegypti*, co-occurrence, pH, season showed significant relationship with larval abundance. The top model explaining *Ae. aegypti* abundance included non-linear relationship (B-spline fit) with pH, and a decrease from February to March and October to December (S8 Table). The sympatry with other species showed no effect on *Ae. aegypti* larval abundance. The top models explaining *Ae. albopictus* larval abundance showed that water volume of breeding habitat as a negative predictor (S9 Table).

## Abiotic factors and urbanization drive niche conservatism in *Aedes* species

Microhabitat sharing between *Ae. aegypti* and *Ae. albopictus* showed significant positive association with pH (OR = 0.15, CI = 0.03–0.68, Wald's $\chi^2$ = 4.2, $p$<0.01).

## How does wing length (as a proxy for body size) vary between interspecific and intraspecific environments?

A total of 235 wings of *Ae. aegypti* (males = 117, females = 119) and 122 wings of *Ae. albopictus* (males = 38, females = 84) were measured. For both larval and adult sampled *Aedes* species, female wing length was longer than male in *Ae. aegypti* (KW $\chi^2$ = 116.32, df = 1, $p$< 0.001) and

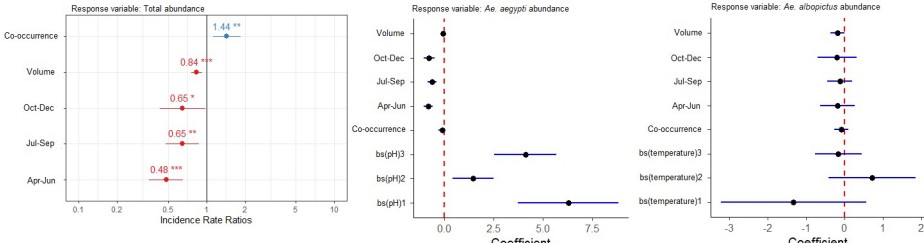

**Fig 9. Model summary of parameter estimates from the best-fit generalized linear mixed-effects model predicting larval abundance, based on a candidate set model combinations of the variables: macrohabitat, microhabitat, pH, temperature, volume, season, and sympatric species.** The models also include the random effects of grid.

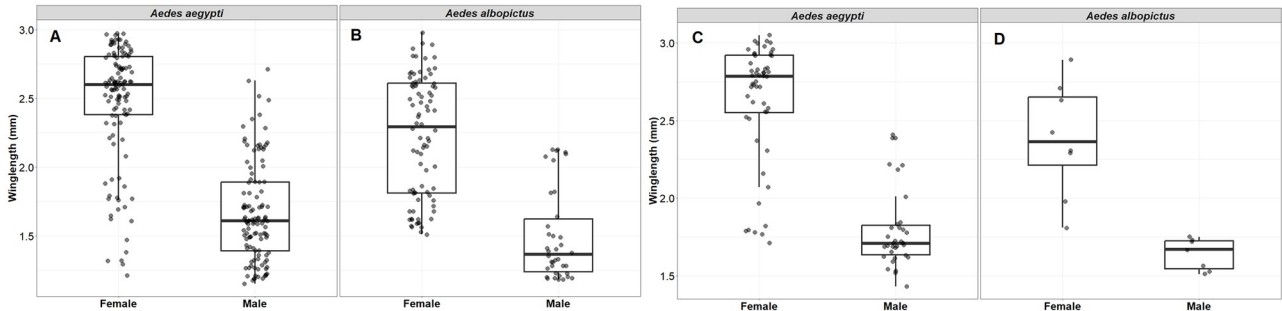

**Fig 10. Comparison of wing length in male and female *Aedes* species as a proxy for body size.** A-B: *Ae. aegypti* and *Ae. albopictus* wing lengths measured from larva emerged populations. C-D: *Ae. aegypti* and *Ae. albopictus* wing lengths of male and female captured as adults using BG traps.

*Ae. albopictus* (KW $\chi^2$ = 52.18, df = 1, $p$< 0.001; Fig 10). Comparisons between wing lengths of adult caught (n = 52) *Ae. aegypti* and larval originated females showed the *Ae. aegypti* mosquitoes caught as adult had longer wing lengths than larval reared females (KW $\chi^2$ = 8.89, df = 1, $p$< 0.002). There was no significant difference in wing lengths of *Ae. albopictus* in adult traps (n = 8) and larva reared (n = 84) KW $\chi^2$ = 0.90, df = 1, $p$ >0.34).

In larval samplings, we measured the effect of sympatry with other *Aedes* species in the breeding sources on the wing length of *Ae. aegypti* and *Ae. albopictus*. The wing length of *Ae. aegypti* showed no significant association with the presence of *Ae. albopictus* (KW $\chi^2$ = 0.83, df = 1, $p$>0.35). However, female *Aedes albopictus* wing lengths were significantly shorter in pools with *Ae. aegypti* (KW $\chi^2$ = 13.21, df = 1, $p$< 0.0002; Fig 11).

The LMM model indicated that pH had a significant negative association with *Ae. aegypti* wing length ($\chi^2$ = 6.54, $p$>0.01) which means female wing length decreased with increase in

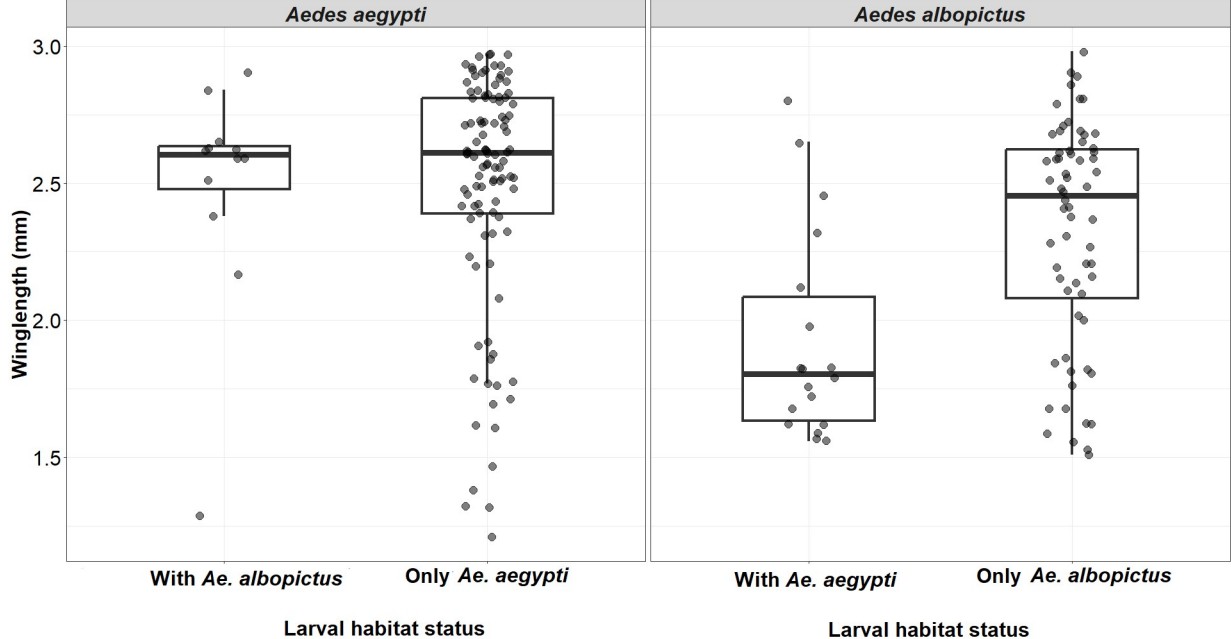

**Fig 11. Wing length comparisons between *Ae. aegypti* and *Ae. albopictus* with and without co-occurrence in same microhabitat.**

pH in larval habitat. In contrast, pH showed positive association with wing length of female *Ae. albopictus*.

## Discussion

Understanding the ecology of mosquito-borne diseases needs a multidimensional and multi-scale approach which can connect fine-scale traits with behaviour, environmental heterogeneity, and epidemiology for realistic predictions of disease dynamics [5]. Cities are constantly perceived as hotspots for infectious diseases [48]. A combination of heterogeneous landscape, population density and socio-ecological drivers such as urbanization, change in land use, and climate change is having indirect effects on the transmission dynamics of infectious diseases. Therefore, combining studies with landscape ecology, microclimate and species composition are essential for designing and monitoring vector control methods. In this study we explored the effects of the prevalence of larval habitat, microclimate, urbanization and traditional *Stegomyia* indices on the larval abundance, body size, species diversity in urban environments.

Our neighbourhood non-residential larval survey showed that *Ae. aegypti* and *Ae. albopictus* were the highly abundant mosquito species which was reflected in the *Stegomyia* indices estimated above the WHO thresholds. Our results show that the container index, Breteau location index and location index showed a linear increase from Barren habitat to High density habitat and significantly high indices from July to September. These findings further support that dengue transmission risk varies even in a heterogenous urbanised landscape and *Aedes* habitat is prevalent in densely populated areas of the city. The increase in *Stegomyia* indices corresponds to seasonal rise in larval prevalence which mirrored the increase in dengue cases in the city (Fig 5). The annual surge in dengue cases starts from early June with the onset of pre-monsoon showers which leads to rise in cases between July and September. Our study relies on outdoor surveillance which highlights the importance of 'neighbourhood surveillance' in public places which can help in real-time forecasting of dengue cases in urban areas [49,50]. There is a little evidence on quantifiable associations between *Stegomyia* indices and dengue transmission that could be reliably used for dengue outbreak prediction [51]. We have no evidence on how the *Stegomyia* indices from indoor surveillance corresponds to dengue outbreaks in the city. There is a need for standardised sampling protocols and well-designed studies that can elucidate the relationship between vector abundance and spatial heterogeneity in dengue transmission. In addition, dengue virus epidemiology is tightly linked to serotype/genotype replacement at the population level which occurs every 2–4 years [52–54]. Bengaluru has experienced triennial peaks, a decline in the total number of dengue cases from 10,411 in 2019 to 2047 in 2020 and 1641 in 2021 [55]. We need fine-scale data on ecological drivers of disease emergence and to understand the links between mosquito population dynamics and how vertical transmission affects the circulating serotypes in the community.

We described significant variations over time and space in larval sites and distributions of two *Aedes* species. The heterogeneity in macrohabitat types was masked by the man-made microhabitats for *Aedes* species which were ubiquitous across the landscape, playing key ecological roles. Habitat preference between two *Aedes* species appeared to be driven at the microhabitat level. Whilst similar habitat association has been reported in Singapore [56], Brazil [10], Burkina Faso [34] and the USA [29], Cameroon [57], Lakshadweep islands [6], Kolkata [58], and Bengaluru [59], the domestic habitats like discarded grinding stones were the most productive microhabitat in Bengaluru and has been recorded in other southern India cities [60,61]. Both *Aedes* species showed high prevalence in discarded grinding stones and negative association with stagnant water. The larval prevalence of *Ae. aegypti* was particularly positively associated with discarded tires. *Ae. albopictus* showed negative association with storage

containers whereas *Aedes aegypti* was marginally positively associated. Storage containers are actively in use but provide high disturbance ephemeral habitats for *Aedes* species. The prevalence of *Ae. aegypti* larval habitat was positively associated with coconut shells, whereas *Ae. albopictus* was recorded in plant axils and tree holes albeit at low frequency and negatively associated with coconut shells.

The high larval prevalence in artificial breeding sites reflected in higher species richness was in artificial sources than natural breeding sources. Storage containers, stagnant water harboured highest mosquito richness probably due to size and water volume allows species to co-exist in the same habitat. Our study revealed that both species richness and diversity were significantly high in plantations and declined in urbanized areas. Similar patterns have been observed in forest community where presence of diverse habitat types and less disturbance supports a diversity of mosquito species [62].

The larval abundance followed the similar pattern as the prevalence of larval habitat for *Ae. aegypti*. The larval abundance of *Ae. aegypti* showed a hump-shaped patterns with an increase from July-September. However, there was no change in abundance of *Ae. albopictus*. To maximise fitness in variable environments an individual often favours a bet-hedging strategy or risk spreading strategy [63], which involves reduction in annual breeding performance and an increase in adult survival so that reproduction can be attempted over more years. The phenology of *Ae. aegypti* and *Ae. albopictus* in invasive regions where they coexist shows that the egg abundance of *Ae. albopictus* becomes more numerous with the progression of wet season [64,65]. Because *Ae. aegypti* has more desiccation resistant eggs compared to *Ae. albopictus* [66,67] one would expect that the frequency of oviposition on the water by *Ae. albopictus* would be greater than that by *Ae. aegypti*. However, we observed similar prevalence but no change in abundance of *Ae. albopictus*. This warrants a detailed study in Bengaluru, where both species are sympatric, to show how two *Aedes* species optimize fitness in varying environments. Water-holding container type, size, organic matters influence in-water oviposition behaviour in *Aedes* species. *Aedes* species larval abundance was negatively associated with volume of larval habitat. Our results are in contrast with Barrera *et al.* [23] which found a positive association between pupal abundance and water volume in containers stored outside the housing. Several studies conducted indoors [10,68] found water volume in larval habitat as a positive predictor of number of immatures with container size. Outdoor habitats are more prone to ecological and anthropological disturbance and truly constitute an ephemeral breeding habitat for *Aedes* mosquitoes.

Among microclimate variables, we found a negative correlation between *Ae. aegypti* larval abundance and water pH which is consistent with the study conducted in Burkina Faso [34]. Temperature showed a negative marginally significant association with *Ae. albopictus* abundance. In general, we found no difference in temperature and humidity across macrohabitat types. However, the mean temperature recorded in the larval habitat was lower than air temperature which supports larval growth throughout the year. Temperature drives the vector ecology by its effect on mosquito behaviour, survival, extrinsic incubation period (EIP) [69]. Temperature thereby shapes parasite transmission by defining lower and upper thermal limits which maintains a non-linear relationship between temperature and length of parasite development [70,71]. The EIP is an important determinant of the temporal dynamics of dengue virus transmission [72]. Rohani *et al.* [73] have reported that the EIP decreases when the extrinsic incubation temperature increases from 9 days at 26˚C to 5 days at 30˚C. We recorded warmer larval habitat temperature (26˚C-28˚C) than air temperature which could affect incubation period of viruses and longer transmission window.

In a landscape context, urbanization was associated with high prevalence of *Aedes* larval habitat and a predominance of artificial containers as breeding sites, mostly colonized by *Ae.*

*aegypti* and *Ae. albopictus*. We found cement tanks used for irrigation purposes infested with mosquito larvae and discarded grinding stones as breeding grounds in plantations and lake areas with no housing and discarded tires as viable larval sites for both *Ae. aegypti* and *An. stephensi*. The high prevalence of artificial habitats contributing to larval prevalence suggests that dengue outbreaks might not only be associated with the biophysical properties (e.g., absence or inaccessibility of piped water supplies or irregular functioning of these piped water systems), but household waste is playing a significant role in driving the *Aedes* ecology and distribution in an urban system. Besides *Aedes* species, *Cx. quinquefasciatus*, *Cx. gelidus*, *Cx. bitaeniorhynchus*, *Cx. mimeticus*, *Ar. subalbatus*, *An. stephensi* and *An. culicifacies* were mainly associated with stagnant water. Therefore, systematic removal of discarded household containers as potential standing water sources is integral to reduce mosquito abundance and ecologically sound method to control arbovirus transmission in Bengaluru city.

Quality of larval habitat, abundance and microclimate impact mosquito body size which has huge epidemiological implications for the carry-over effect of the immature mosquito's life on *Ae. aegypti* competence for arbovirus transmission [74]. *Aedes* body size showed a variation in wing length across microhabitat and macrohabitat types. The wing length of mosquitoes that emerged from the discarded grinding stones were significantly larger than discarded containers or coconut shells. The best model showed a negative association between pH and wing length for *Ae. aegypti* and a contrasting effect on *Ae. albopictus* with positive association between pH and wing length. Small body size females cannot fly long-distance and require multiple meals whereas large body size females can fly farther. Co-existence with other species seems to affect only *Ae. albopictus* with reduction in wing length. This was in contrast with study conducted in Brazil [10]. Interspecific larval competition between *Ae. aegypti* and *Ae. albopictus* in laboratory study have shown greater effect on survival and wing length of *Ae. albopictus*, at intermediate larval density [75].

## Conclusions

Our longitudinal study provides insights into microhabitat and macrohabitat of two dengue vectors i.e., *Ae. aegypti* and *Ae. albopictus* in outdoor survey across heterogeneous landscapes. Mosquito species diversity was higher in plantation areas than in densely populated grids. However, *Aedes* infestation indices were highest in urban grids between July and September. We found artificial habitats such as discarded grinding stones and storage containers are the major contributors for mosquito larval habitat. This further implies that improper storage practices and unplanned waste management in the city leads to vector breeding. We provide a robust understanding on the ecological drivers of mosquito habitat that will help in efficient vector control and implementing innovative strategies.

## Supporting information

**S1 Table. *Aedes* (*Stegomyia*) infestation indices across macrohabitat types.**
(XLSX)

**S2 Table. Mosquito larval habitat prevalence by microhabitat types.**
(XLSX)

**S3 Table. TukeyHSD posthoc test for comparison across microhabitat types.**
(XLSX)

**S4 Table. Candidate models to predict total larval prevalence.**
(XLSX)

**S5 Table. Candidate models to predict *Ae. aegypti* larval prevalence.**
(XLSX)

**S6 Table. Candidate models to predict *Ae. albopictus* larval prevalence.**
(XLSX)

**S7 Table. Candidate models to predict Total larval abundance.**
(XLSX)

**S8 Table. Candidate models to predict *Ae. aegypti* larval abundance.**
(XLSX)

**S9 Table. Candidate models to predict *Ae. albopictus* larval abundance.**
(XLSX)

**S1 Fig. Mean temperature across macrohabitats.**
(TIF)

**S1 Data. Prevalence and abundance of mosquito larvae by macro and microhabitat across index and random grids.**
(XLSX)

## Acknowledgments

We would like to thank the Bruhat Bengaluru Mahanagara Palike (BBMP) for permission and support for this study. We would like to thank Sutharsan G. for help in initial stages of the field work.

## Author Contributions

**Conceptualization:** Farah Ishtiaq.

**Data curation:** Deepa Dharmamuthuraja, Rohini P. D., Iswarya Lakshmi M., Farah Ishtiaq.

**Formal analysis:** Deepa Dharmamuthuraja, Rohini P. D., Farah Ishtiaq.

**Funding acquisition:** Farah Ishtiaq.

**Investigation:** Susanta Kumar Ghosh, Farah Ishtiaq.

**Methodology:** Kavita Isvaran, Susanta Kumar Ghosh, Farah Ishtiaq.

**Project administration:** Deepa Dharmamuthuraja, Rohini P. D., Farah Ishtiaq.

**Resources:** Farah Ishtiaq.

**Software:** Farah Ishtiaq.

**Supervision:** Farah Ishtiaq.

**Validation:** Farah Ishtiaq.

**Visualization:** Deepa Dharmamuthuraja, Rohini P. D., Farah Ishtiaq.

**Writing – original draft:** Deepa Dharmamuthuraja, Farah Ishtiaq.

**Writing – review & editing:** Kavita Isvaran, Susanta Kumar Ghosh, Farah Ishtiaq.

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
