## [Decision Letter · Decision Letter 0]

28 Aug 2023

Dear Dr. Ishtiaq,

Thank you very much for submitting your manuscript "Determinants of Aedes mosquito larval ecology in a heterogeneousurban environment- a longitudinal study in Bengaluru, India" for consideration at PLOS Neglected Tropical Diseases. As with all papers reviewed by the journal, your manuscript was reviewed by members of the editorial board and by several independent reviewers. The reviewers appreciated the attention to an important topic. Based on the reviews, we are likely to accept this manuscript for publication, providing that you modify the manuscript according to the review recommendations. 

Sincerely,

Roberto Barrera, Ph.D.

Academic Editor

Audrey Lenhart

Section Editor

Reviewer's Responses to Questions

**Key Review Criteria Required for Acceptance?**

**Methods**

-Are the objectives of the study clearly articulated with a clear testable hypothesis stated?

-Is the study design appropriate to address the stated objectives?

-Is the population clearly described and appropriate for the hypothesis being tested?

-Is the sample size sufficient to ensure adequate power to address the hypothesis being tested?

-Were correct statistical analysis used to support conclusions?

-Are there concerns about ethical or regulatory requirements being met?

Reviewer #1: The methods used for this study are highly appropriate with an excellent use of mixed effect models; an appropriate way of determining key drivers of mosquito prevalence. The authors used models to inform and guide rather than provide a definitive set of predictors which is a welcome approach to such analysis.

The methods are laid out well and clear for the most part, although the only minor comments are:

Line 200: a reference to the minimum distance between each grid could be made clearer – which grids are being referred to, either the Index grid or Random grid? I assume the Random grids, but it would be worth being explicit for the reader.

Lines 201 – 204: It would be worth stating how long the ibuttons ran for. The whole of the sampling period or just one/two/more days? Likewise, it isn’t clear whether the ibuttons were deployed in the Index or Random grids, or indeed, both.

Reviewer #2: Line 173: The phrase that starts with "September being" is not a complete sentence.

Line 184: Culicidae mosquito is redundant.

Lines 190, 191, 192 and thereafter: density, not dense

Line 234: mosquitoes (plural)

Were data tested for normality prior to ANOVA?

Were the indices used in these calculations invented by the authors or were they derived from literature?

I like that the authors measured temperatures for both air and water inside container habitats. This is an often-overlooked criterion.

For all statistical tests that are used, please include a short statement about reporting of results, for example, for ANOVA, "Results are reported as arithmetic mean +/- standard error." This gives the reader an idea of what to expect.

**Results**

-Does the analysis presented match the analysis plan?

-Are the results clearly and completely presented?

-Are the figures (Tables, Images) of sufficient quality for clarity?

Reviewer #1: The authors analysed the data completely as planned, and present a clear set of results.

There is a well presented data overview, covering core data before moving onto more complex analysis; each statistical analysis was correctly presented.

With the mixed model analysis, the authors kept to the key drivers/main models whilst presenting their results - I fully approve of this approach, as there is a tendancy to over-describe results from mixed models which can muddy the waters somewhat. Because of this approach, the results, particularly for the mixed effect models was clear and easy to follow.

Figures are of good quality and used to emphasise main findings. As such, the figures complement the writing and help enhance the narrative.

Reviewer #2: Lines 371-373: Were these species mentioned in full prior to this paragraph? I must have missed that.

I see chi-square values reported but I didn't find them mentioned in the methods section. Same with beta values and Chao1 values.

Line 523: This is not a complete sentence. What am I supposed to understand from a half of a thought?

Lines 387-388: the authors mention no seasonal difference between BPR and they refer to Figure 3, but figure 3 doesn't give any information about seasonality of BPR.

Line 394: reference for WHO thresholds?

**Conclusions**

-Are the conclusions supported by the data presented?

-Are the limitations of analysis clearly described?

-Do the authors discuss how these data can be helpful to advance our understanding of the topic under study?

-Is public health relevance addressed?

Reviewer #1: The study has clear relevance for informing disease control from mosquito vectors. I completely concur with the conclusions drawn from the data. 

The influence of the environment on mosquito breeding success, particularly those species linked with Dengue transmission, is clearly an important step in helping to control disease outbreaks. The study covered microhabitat and macrohabitat drivers and found that microhabitat, notably container presence (as well as season) helped determine mosquito presence and larval prevalence. This prevalence increased in urbanised areas, whilst species diversity decreased in urbanised areas leaving Aedes species as dominant species, clearly linked to dengue transmission. 

The authors correctly state that "transmission reduction programmes should focus on ‘neighbourhood surveillance’" and within this context, reduction or removal of containers in urbanised areas would be a principal mechanism of reducing Aedes-borne diseases. The authors recognise that the link between dengue outbreaks and mosquito population dynamics is not clear, therefore further fine-scale data is required.

I can safely say that the relevance to public health is very clearly addressed.

Reviewer #2: I believe that the authors have drawn reasonable conclusions based on their work.

**Editorial and Data Presentation Modifications?**

Reviewer #1: As outlined above:

Line 200: a reference to the minimum distance between each grid could be made clearer – which grids are being referred to, either the Index grid or Random grid? I assume the Random grids, but it would be worth being explicit for the reader.

Lines 201 – 204: It would be worth stating how long the ibuttons ran for. The whole of the sampling period or just one/two/more days? Likewise, it isn’t clear whether the ibuttons were deployed in the Index or Random grids, or indeed, both.

Reviewer #2: Some of the figures are a little hard to see - detail is lost at the small scale. I wonder what the figures will look like in print. I hope that they are readable.

**Summary and General Comments**

Reviewer #1: I felt that this study was really well conducted, thought-out and analysed to a high standard. The relevance to public health is clear and with all studies obvioulsy drives a number of additional questions. The simplicity of the conclusions is fantastic - something that can be implemented at ground-level. It will be interesting to see how that message is communicated from the scientific literature to the relevant neighborhoods; I fear that communicating the message will be a harder task than the study.

Reviewer #2: In the abstract, Line 78, Aedes albopictus =Stegomyia albopicta. Aedes and albopictus are masculine but Stegomyia is feminine and so the ending must agree, ergo albopicta.

PLOS authors have the option to publish the peer review history of their article (what does this mean?). If published, this will include your full peer review and any attached files.

Reviewer #1: Yes: Michael Bungard

Reviewer #2: No

Figure Files:

Data Requirements:

Reproducibility:

References

---

## [Editor Report · Decision Letter 1]

5 Oct 2023

Dear Dr. Ishtiaq,

We are pleased to inform you that your manuscript 'Determinants of Aedes mosquito larval ecology in a heterogeneous urban environment- a longitudinal study in Bengaluru, India' has been provisionally accepted for publication in PLOS Neglected Tropical Diseases.

Best regards,

Roberto Barrera, Ph.D.

Academic Editor

Audrey Lenhart

Section Editor

---

## [Editor Report · Acceptance letter]

1 Nov 2023

Dear Dr. Ishtiaq,

We are delighted to inform you that your manuscript, "Determinants of Aedes mosquito larval ecology in a heterogeneous urban environment- a longitudinal study in Bengaluru, India," has been formally accepted for publication in PLOS Neglected Tropical Diseases.

Best regards,

Shaden Kamhawi

co-Editor-in-Chief

Paul Brindley

co-Editor-in-Chief
